# Unsupervised Adaptation for Fairness under Covariate Shift

## Abstract

Training fair models typically involves optimizing a composite objective accounting for both prediction accuracy and some fairness measure. However, due to a shift in the distribution of the covariates at test time, the learnt fairness tradeoffs may no longer be valid, which we verify experimentally. To address this, we consider an unsupervised adaptation problem of training fair classifiers when only a small set of unlabeled test samples is available along with a large labeled training set. We propose a novel modification to the traditional composite objective by adding a weighted entropy objective on the unlabeled test dataset. This involves a min-max optimization where weights are optimized to mimic the importance weighting ratios followed by classifier optimization. We demonstrate that our weighted entropy objective provides an upper bound on the standard importance sampled training objective common in covariate shift formulations under some mild conditions. Experimentally, we demonstrate that Wasserstein distance based penalty for representation matching across protected sub groups together with the above loss outperforms existing baselines. Our method achieves the best accuracy-equalized odds tradeoff under the covariate shift setup. We find that, for the same accuracy, we get up to $2\times$ improvement in equalized odds on notable benchmarks.

## 1 Introduction

Moving away from optimizing only prediction accuracy, there is a lot of interest in understanding and analyzing Machine Learning model performance along other dimensions like robustness (Silva & Najafirad, 2020), model generalization (Wiles et al., 2021) and fairness (Oneto & Chiappa, 2020). In this work, we focus on the algorithmic fairness aspect. When the prediction of a machine learning classifier is used to make important decisions that have societal impact, like in criminal justice, loan approvals, to name a few; how decisions impact different protected groups needs to be taken into account. Datasets used for training could be biased in the sense that some groups may be under-represented, biasing classifier decisions towards the over-represented group or the bias could be in terms of undesirable causal pathways between sensitive attribute and the label in the real world data generating mechanism (Oneto & Chiappa, 2020). It has often been observed (Bolukbasi et al., 2016), (Buolamwini & Gebru, 2018) that algorithms that optimize predictive accuracy that are fed pre-existing biases further learn and then propagate the same biases.

While there are various approaches for fair machine learning, a class of methods called in-processing methods have been shown to perform well (Wan et al., 2021). These methods regularize training of fair models typically through a composition of loss objective accounting for a specific fairness measure along with predictive accuracy. Popular fairness measures are based on notions of demographic parity, equal opportunity, predictive rate parity and equalized odds. After regularized training, the model attains a specific fairness-accuracy tradeoff. When the test distribution is close or identical to the training distribution, fairness-accuracy tradeoffs typically hold. However, in practical scenarios, there could be non-trivial distributional shifts due to which tradeoffs achieved in train may not hold in the test. For example, Ding et al. (2021) highlights how a classifier's fairness-accuracy tradeoff trained on input samples derived from one state does not extend to predict income in other states for the Adult Income dataset. Similarly, Rezaei et al. (2021); Mandal et al. (2020) demonstrate that the tradeoffs achieved by state of the art fairness techniques do not generalize to test data under shifts. In figure 1, we complement these claims by analyzing the *under-performance* for a state-of-the-art

fairness method - Adversarial Debiasing (Zhang et al., 2018). We also see similar drop in performance under covariate shift in other baselines we consider, which we highlight in our experimental analysis.

In this work, we study covariate shift where the distribution of covariates across training and testing changes, however the optimal label predictor conditioned on input remains the same. We address the following question for unsupervised adaptation of training fair classifiers: *Under the covariate shift setup, given sufficient amount of labeled training samples and only a few unlabeled test samples, how can we ensure good fairness-accuracy trade-offs on the test distribution?*

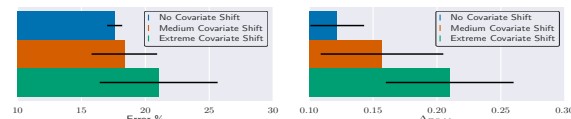

Figure 1: Both Error (in % left) and Equalized Odds (right) for SOTA fairness method - Adversarial Debiasing exhibit strong degradation on increasing the magnitude of covariate shift. Three scenarios corresponding to no shift, intermediate shift and high shift are plotted (details on shift construction are provided in experiments).

While this question has not received much attention, some recent works like Rezaei et al. (2021) have begun to address this problem. Prior works rely on explicit density estimation which is then used to adapt to test data. In our work, we focus on avoiding density estimation steps that do not scale well in high dimensions. Here, we propose a novel unsupervised adaptation training objective that is theoretically justified. The objective depends on labeled training samples and unlabeled test samples along with standard fairness objective involving representation matching across the groups on the test. We report the results on *equalized odds* in our experiments and use the related notion of *accuracy parity* to motivate our algorithmic design with empirical evidence. Our key contributions are listed as follows:

1. We show that under a scenario of *asymmetric covariate shift*, where one group exhibits large covariate shift while the other does not, accuracy parity degrades despite perfect representation matching across protected groups highlighting the need to tackle covariate shift explicitly. (Section 4)

2. We introduce a composite objective for prediction that involves *a novel weighted entropy objective on the set of unlabeled test samples* along with standard a ERM objective on the labeled training samples for tackling covariate shift. We optimize the weights using *min-max* optimization: The outer minimization optimizes the classifier with the composite objective, while the inner maximization finds the appropriate weights for each sample that are related to importance sampling ratios determined *implicitly* with no density estimation steps. We prove that our composite objective provides an upper bound on the standard importance sampled training objective common in covariate shift formulations under some mild conditions. We then combine the above composite objective with a representation matching loss to train fair classifiers. (Section 5)

3. We experiment on four benchmark datasets, including Adult, Arrhythmia, Communities and Drug. We demonstrate that, by incorporating our proposed weighted entropy objective, with the Wasserstein based penalty for representation matching across protected sub-groups, we outperform existing fairness methods under covariate shifts. In particular, we achieve the best accuracy-equalized odds tradeoff: for the same accuracy, we achieve up to $\approx 2\times$ improvement in equalized odds metric. (Section 6)

## 2   RELATED WORK

**Fairness Metrics:** There have been works studying different types of fairness criterion. *Group* Fairness metrics have been studied in Hardt et al. (2016b); Kleinberg et al. (2016) while *Individual* Fairness metrics were studied in Dwork et al. (2012); Sharifi-Malvajerdi et al. (2019), *Causal* Fairness criterions has been studied in Kilbertus et al. (2017); Kusner et al. (2017); Galhotra et al. (2022); Chiappa (2019); Nabi et al. (2019); Salimi et al. (2019) where causal mechanisms that generate data are leveraged. Our work addresses questions surrounding statistical Group Fairness metrics where we address the effect of covariate shift on fairness-accuracy tradeoffs.

**Techniques for imposing fairness:** *Pre-processing* techniques that aim to transform the dataset (Calmon et al., 2017; Swersky et al., 2013; Feldman et al., 2015; Kamiran & Calders, 2012) followed by a standard training have been studied. *In-processing* methods directly modify the learning

algorithms using techniques, such as, adversarial learning (Madras et al., 2018; Zhang et al., 2018), (Agarwal et al., 2018; Cotter et al., 2019; Donini et al., 2018; Fish et al., 2016; Zafar et al., 2017; Celis et al., 2019). *Post-processing* approaches, primarily focus on modifying the outcomes of the predictive models in order to make unbiased predictions (Pleiss et al., 2017; Zhao et al., 2017; Hardt et al., 2016b). Bellamy et al. (2019) provides a comprehensive survey containing a broad variety of these algorithms. Our method is an in-processing technique and is different from the above methods in that it operates with a small unlabeled test set along with a standard labeled training set.

**Distribution Shift:** Research addressing distribution shift in machine learning is vast and is growing. The general case considers a joint distribution shift between training and testing data (Ben-David et al., 2006; Blitzer et al., 2007; Moreno-Torres et al., 2012) resulting in techniques like domain adaptation (Ganin & Lempitsky, 2015), distributionally robust optimization (Sagawa et al., 2019; Duchi & Namkoong, 2021) and invariant risk minimization and its variants (Arjovsky et al., 2019; Krueger et al., 2021; Shi et al., 2021). A survey of various methods and their relative performance is discussed by Wiles et al. (2021). We focus on the problem of *Covariate Shift* where the *Conditional Label* distribution is invariant while there is a shift in the marginal distribution of the covariates across training and test samples. This classical setup is studied by Shimodaira (2000); Sugiyama et al. (2007b); Gretton et al. (2009). *Importance Weighting* is one of the prominently used techniques for tackling covariate shifts (Sugiyama et al., 2007a; Lam et al., 2019). However, they are known to have high variance under minor shift scenarios (Cortes et al., 2010a). Recently methods that emerged as the de-facto approaches to tackle distribution shifts include popular entropy minimization (Wang et al., 2021a), pseudo-labeling (French et al., 2017; Xie et al., 2020), batch normalization adaptation (Schneider et al., 2020; Nado et al., 2020), because of their wide applicability and superior performance. Our work provides a connection between a version of weighted entropy minimization and traditional importance sampling based loss which may be of independent interest.

**Fairness under Distribution shift:** The work by Rezaei et al. (2021) is by far the most aligned to ours as they propose a method that is robust to covariate shift while ensuring fairness when unlabeled test data is available. However, this requires the density estimation of training and test distribution that is not efficient at higher dimensions and small number of test samples. In contrast our method avoids density estimation and uses a weighted version of entropy minimization that is constrained suitably to reflect importance sampling ratios implicitly. Mandal et al. (2020) proposed a method for fair classification under the worst-case weighting of the data via an iterative procedure, but it is in the agnostic setting where test data is not available. Singh et al. (2021) studied fairness under shifts through a causal lens but the method requires access to the causal graph, separating sets and other non-trivial data priors. Zhang et al. (2021) proposed FARF, an adaptive method for learning in an online setting under fairness constraints, but is clearly different from the static shift setting considered in our work. Slack et al. (2020) proposed a MAML based algorithm to learn under fairness constraints, but it requires access to labeled test data. An et al. (2022) propose a consistency regularization technique to ensure fairness under label shifts, while we consider covariate shift.

## 3 PROBLEM SETUP

Let $\mathcal{X} \subseteq \mathcal{R}^d$ be the $d$ dimensional feature space for covariates, $\mathcal{A}$ be the space of categorical *group* attributes and $\mathcal{Y}$ be the space of class labels. In this work, we consider $\mathcal{A} = \{0, 1\}$ and $\mathcal{Y} = \{0, 1\}$. Let $\mathbf{X} \in \mathcal{X}, \mathbf{A} \in \mathcal{A}, \mathbf{Y} \in \mathcal{Y}$ be realizations from the space. We consider a training dataset $\mathcal{D}^S = \{(\mathbf{X}_i, \mathbf{A}_i, \mathbf{Y}_i) | i \in [n]\}$ where every tuple $(\mathbf{X}_i, \mathbf{A}_i, \mathbf{Y}_i) \in \mathcal{X} \times \mathcal{A} \times \mathcal{Y}$. We also have an *unlabeled* test dataset, $\mathcal{D}^T = \{\mathbf{X}_i, \mathbf{A}_i | i \in [m]\}$. We focus on the setup where $m << n$. The training samples $(\mathbf{X}_i, \mathbf{A}_i, \mathbf{Y}_i \in \mathcal{D}^S)$ are sampled i.i.d from distribution $\mathbb{P}^S(\mathbf{X}, \mathbf{Y}, \mathbf{A})$ while the unlabeled test instances are sampled from $\mathbb{P}^T(\mathbf{X}, \mathbf{A})$.

Let $\mathcal{F} : \mathcal{X} \to [0, 1]$ be the space of soft prediction models. In this work, we will consider $F \in \mathcal{F}$ of the form $F = h \circ g$ where $g(\mathbf{X}) \in \mathbb{R}^k$ (for some dimension $k > 0$), is a representation that is being learnt while $h(g(\mathbf{X})) \in [0, 1]$ provides the soft prediction. Note that we don't consider A as an input to $F$, as explained in the work of (Zhao, 2021). The parameters of $F$ are denoted as $\theta(F)$. We denote the class prediction probabilities from $F$ with $P(\hat{Y} = y | \mathbf{X}_i)$, where $y \in \{0, 1\}$.

The supervised in-distribution training of $F$ is done by minimizing the *empirical risk*, $\widehat{ER}^S$ as the proxy for *population risk*, $\mathcal{R}^S$. Both risk measures are computed using the *Cross Entropy (CE)* loss

for classification (correspondingly we use $\widehat{ER}^T$ and $\mathcal{R}^T$ over the *test distribution* for $F$).

$$\mathcal{R}^S = \mathbb{E}_{\mathbb{P}^S(\mathbf{X},\mathbf{A},\mathbf{Y})}\left(-\log P(\hat{\mathbf{Y}} = \mathbf{Y}|\mathbf{X})\right), \ \widehat{ER}^S = \frac{1}{n}\sum_{(\mathbf{X}_i,\mathbf{Y}_i,A_i)\in\mathcal{D}^S}\left(-\log P(\hat{\mathbf{Y}} = \mathbf{Y}_i|\mathbf{X}_i)\right)$$

(1)

### 3.1 COVARIATE SHIFT ASSUMPTION

For our work, we adopt the *covariate shift* assumption as in Shimodaira (2000). Covariate shift assumption implies that $\mathbb{P}^S(\mathbf{Y}|\mathbf{X},\mathbf{A}) = \mathbb{P}^T(\mathbf{Y}|\mathbf{X},\mathbf{A})$. In other words, shift in distribution only affects the joint distribution of covariates and sensitive attribute, i.e. $\mathbb{P}^S(\mathbf{X},\mathbf{A}) \neq \mathbb{P}^T(\mathbf{X},\mathbf{A})$. We note that our setup is identical to a recent work of fairness under covariate shift by Rezaei et al. (2021). We also define and focus on a special case of covariate shift called *asymmetric covariate shift*.

**Definition 1** (Asymmetric Covariate Shift). Asymmetric covariate shift occurs when distribution of covariates of one group shifts while the other does not, i.e. $\mathbb{P}^T(\mathbf{X}|A = 1) \neq \mathbb{P}^S(\mathbf{X}|A = 1)$ while $\mathbb{P}^T(\mathbf{X}|A = 0) = \mathbb{P}^S(\mathbf{X}|A = 0)$ in addition to $\mathbb{P}^S(\mathbf{Y}|\mathbf{X},\mathbf{A}) = \mathbb{P}^T(\mathbf{Y}|\mathbf{X},\mathbf{A})$

This type of covariate shift occurs when a sub-group is over represented (sufficiently capturing all parts of the domain of interest in the training data) while the other sub-group being under represented and observed only in one part of the domain. In the test distribution, covariates of the under-represented group assume a more drastic shift.

### 3.2 FAIRNESS MEASURE

To quantify fairness, we follow Rezaei et al. (2021) and use *Equalized Odds (EOdds)*, proposed by Hardt et al. (2016a): $\Delta_{\text{EOdds}} = \frac{1}{2}\sum_{y\in\{0,1\}}|P(\hat{\mathbf{Y}} = 1|A = 0, \mathbf{Y} = y) - P(\hat{\mathbf{Y}} = 1|A = 1, \mathbf{Y} = y)|$. EOdds requires parity in both true positive rates and false positive rates across the groups. Hardt et al. (2016a) have raised several concerns regarding other widely used fairness metrics, e.g., Demographic Parity (DP) and Equalized Opportunity (EOpp). Therefore, we don't emphasize them in this work. Another way to interpret EOdds is that it requires $I(\hat{\mathbf{Y}}; A|\mathbf{Y})$ to be small, where $I(; |\cdot)$ is the *conditional mutual information* measure. Ideally, we are interested in a classifier, $F$ that minimizes the objective: $\mathcal{R}^T + \lambda I_T(\hat{\mathbf{Y}}; A|\mathbf{Y})$; where $I_T(\cdot)$ is the mutual information measure with respect to the test distribution. However, EOdds metric requires the true labels $Y$ from the test distribution. Therefore, we consider optimizing for a related weaker notion, called *accuracy parity*, i.e. $\Delta_{\text{Apar}} = |P(\hat{\mathbf{Y}} \neq \mathbf{Y}|A = 0) - P(\hat{\mathbf{Y}} \neq \mathbf{Y}|A = 1)|$. In information theoretic terms, minimizing accuracy parity entails keeping $I_T(\hat{\mathbf{Y}} \neq Y; A)$ small. We now state the main goal of this work:

**Objective** $$\min_{\theta(F)} \mathcal{R}^T + \lambda\Delta_{\text{Apar}}.$$ (2)

## 4 REPRESENTATION MATCHING AND COVARIATE SHIFT

Our objective is to learn a highly accurate classifier on the test distribution while ensuring accuracy parity as in (2). Despite the lack of test labels, accuracy parity admits a simpler sufficient condition: Train a classifier $F = h \circ g(\mathbf{X})$ by matching representation $g(\mathbf{X})$ across the protected sub groups and learning a classifier on top of that representation (Zhao & Gordon, 2019). Several variants for representation matching loss have been proposed in the literature for both classification (Jiang et al., 2020; Wang et al., 2021c) and regression (Zhao, 2021; Chzhen et al., 2020). For implementation ease, we pick Wasserstein-2 metric to impose representation matching. We recall the definition of Wasserstein distance:

**Definition 2.** Let $(\mathcal{M}, d)$ be a metric space and $P_p(\mathcal{M})$ denote the collection of all probability measures $\mu$ on $\mathcal{M}$ with finite $p^{th}$ moment. Then the $p$-th Wasserstein distance between measures $\mu$ and $\nu$ both $\in P_p(\mathcal{M})$ is given by: $\mathcal{W}_p(\mu,\nu) = \left(\inf_\gamma \int_{\mathcal{M}\times\mathcal{M}} d(x,y)^p d\gamma(x,y)\right)^{\frac{1}{p}}$; $\gamma \in \Gamma(\mu,\nu)$, where $\Gamma(\mu,\nu)$ denotes the collection of all measures on $\mathcal{M}\times\mathcal{M}$ with marginals $\mu$ and $\nu$ respectively.

We minimize the $\mathcal{W}_2$ between the representation $g(\cdot)$ of the test samples from both groups. Empirically, our representation matching loss is given by:

$$\hat{\mathcal{L}}_{Wass}(\mathcal{D}^T) = \mathcal{W}_p(\hat{\mu}, \hat{\nu}), \ \hat{\mu} = \frac{\sum_{(\mathbf{X}_i, \mathbf{A}_i=0) \in \mathcal{D}^T} \delta_{g(\mathbf{X}_i)}}{|(\mathbf{X}_i, \mathbf{A}_i = 0) \in \mathcal{D}^T|}, \ \hat{\nu} = \frac{\sum_{(\mathbf{X}_i, \mathbf{A}_i=1) \in \mathcal{D}^T} \delta_{g(\mathbf{X}_i)}}{|(\mathbf{X}_i, \mathbf{A}_i = 1) \in \mathcal{D}^T|} \tag{3}$$

We arrive at the following objective:

$$\min_{\theta(F=h \circ g)} \widehat{ER}^T + \lambda \hat{\mathcal{L}}_{Wass}(\mathcal{D}^T) \tag{4}$$

However, we still don't have labeled samples from $\mathcal{D}^T$ for realizing the first term. It is natural to optimize the following objective: $\widehat{ER}^S + \lambda \hat{\mathcal{L}}_{Wass}(\mathcal{D}^T)$ where the first term leverages labeled training data while the second term matches representation across groups using unlabeled test. We illustrate that under covariate shift, using the test set only for representation matching alone is ineffective. We also provide strong experimental justification to support this claim in section A.6.2. We now quote a result from existing literature that bounds accuracy parity under representation matching.

**Theorem 1** (Zhao & Gordon (2019)). *Consider any soft classifier $F = h \circ g(\mathbf{X}) \in [0, 1]$ and the hard decision rule $\hat{Y} = \mathbf{1}_{F(\mathbf{X})>1/2}$. Let the Bayes optimal classifier for group $a$ under representation $g(\cdot)$ be: $\mathbf{1}_{\mathbb{P}^T(\mathbf{Y}=1|g(\mathbf{X}),A=a)>1/2} = s_a(\mathbf{X})$. Let the Bayes error for group $a$ under representation $g(\cdot)$ be $\mathrm{err}_a$. Then we have:*

$$\Delta_{\mathrm{Apar}} \leq \sum_a \mathrm{err}_a + \|\mathbb{P}^T(g(\mathbf{X})|A=1) - \mathbb{P}^T(g(\mathbf{X})|A=0)\|_1 + \min_a (\mathbb{E}_{\mathbb{P}^T(\mathbf{X}|a)}|s_1(\mathbf{X}) - s_0(\mathbf{X})|)$$

*Here, $\|\mathbb{P}(\cdot) - \mathbb{Q}(\cdot)\|_1$ is the total variation distance between measures $\mathbb{P}$ and $\mathbb{Q}$.*

This suggests applying a loss for representation matching to enforce accuracy parity as it would drive the purely label independent middle term to zero. However, we argue that, under *asymmetric covariate shift* (Definition 1), accuracy parity is approximately the third term in Theorem 1, even when the second term is set to 0.

**Representation Matching does not work under Asymmetric Covariate Shift:** Consider the covariate shift scenario given by Definition 1. Suppose one is also able to find a representation $g(\cdot)$ that matches across groups exactly in the test, i.e. $\mathbb{P}^T(g(\cdot)|A=1) = \mathbb{P}^T(g(\cdot)|A=0)$. Due to the asymmetric covariate shift assumption between train and test, we have $\mathbb{P}^T(g(\cdot)|A=0) = \mathbb{P}^T(g(\cdot)|A=1) = \mathbb{P}^S(g(\cdot)|A=0)$. Since there is no covariate shift for group $A=0$, optimal scoring function $s_0(\mathbf{X})$ remains the same even for the training set, given the representation.

Since a classifier $h$ is learnt on top of representation $g$, and only training distribution of group $A=0$ under $g$ overlaps (completely) with the test, classifier $h$ would be trained overwhelmingly with the correct labels for $A=0$ in the region where test samples are found. Over the test distribution, the hard decision score function will be approximately $s_0(\mathbf{X})$. Therefore, the error in the group 0 would be small. While the test error in group 1 will be approximately $\mathbb{E}_{\mathbb{P}^T(\cdot|A=1)}(|s_0(\mathbf{X}) - s_1(\mathbf{X})|)$ which matches the third term in Theorem 1.

Therefore, in this setting it is essential to use training samples and unlabeled test samples to address covariate shift problem for group 1. Samples for group 1 in the training distribution (not just that belong to group $A=0$) must be emphasized more. This motivates the need for performing unsupervised adaptation using unlabeled test samples focusing on accuracy improvement and combining it with representation matching.

## 5 METHOD AND ALGORITHM

Recall that the objective we are interested in is (4). One needs a proxy for the first term due to lack of labels. From considerations in the previous section, training has to be done in a manner that can tackle covariate shift despite using representation matching. Building over the analysis from the previous section, we derive a novel objective in Theorem 2 based on the weighted entropy over instances in $\mathcal{D}^T$ along with empirical loss over $\mathcal{D}^S$ and show that is an upper bound to $\mathcal{R}^T$.

**Theorem 2.** *Suppose that $\mathbb{P}^T(\cdot)$ and $\mathbb{P}^S(\cdot)$ are absolutely continuous with respect to each other over domain $\mathcal{X}$. Let $\epsilon \in \mathbb{R}^+$ be such that $\frac{\mathbb{P}^T(Y=y|\mathbf{X})}{P(\hat{Y}=y|\mathbf{X})} \leq \epsilon$, for $y \in \{0,1\}$ almost surely with respect to distribution $\mathbb{P}^T(\mathbf{X})$. Then, we can upper bound $\mathcal{R}^T$ using $\mathcal{R}^S$ along with an unsupervised objective over $\mathbb{P}^T$ as:*

$$\mathcal{R}^T \leq \mathcal{R}^S + \epsilon \times \mathbb{E}_{\mathbb{P}^T(\mathbf{X})}\left[e^{\left(-\frac{\mathbb{P}^S(\mathbf{X})}{\mathbb{P}^T(\mathbf{X})}\right)}\mathcal{H}(\hat{Y}|\mathbf{X})\right] \tag{5}$$

*where $\mathcal{H}(\hat{Y}|\mathbf{X}) = \sum_{y\in\{0,1\}} -P(\hat{Y}=y|\mathbf{X})\log(P(\hat{Y}=y|\mathbf{X}))$ is the conditional entropy of the label given a sample $\mathbf{X}$.*

*Proof Sketch.* The proof is relegated to the appendix A.1. To arrive at this bound, we manipulate the importance sampled population training loss (Sugiyama et al., 2007b). □

We *emphasize* that this result also provides an important connection and a rationale for using entropy based objectives as an unsupervised adaptation objective from an importance sampling point of view that has been missing in the literature (Wang et al., 2021a; Sun et al., 2019).

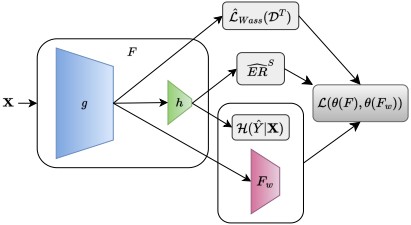

Figure 2: High level architecture of our method. Colored blocks represent parameterized subnetworks.

Entropy objective is imposed on points that are more typical with respect to the test than the training. Conversely, in the region where samples are less likely with respect to the test distribution, since it has been optimized for label prediction as part of training, the entropy objective is not imposed strongly. The above bound however hinges on the assumption that pointwise in the domain $\mathcal{X}$, $F$ approximates the true soft predictor by at most a constant factor $\epsilon$. To ensure a small value of $\epsilon$, we resort to pre-training $F$ with only $\mathcal{D}^S$ samples for a few epochs before imposing any other type of regularization.

### 5.1 WEIGHTED ENTROPY OBJECTIVE

Implementing the objective in (5), requires computation of the Radon-Nikodym derivative $\frac{d\mathbb{P}^S(\mathbf{X})}{d\mathbb{P}^T(\mathbf{X})}$. This is challenging when $m$ (amount of unlabeled test samples) is small and typical way of density estimation in high dimensions is particularly hard. Therefore, we propose to estimate the ratio $\frac{d\mathbb{P}^S(\mathbf{X})}{d\mathbb{P}^T(\mathbf{X})}$ by a parametrized network $F_w : \mathcal{X} \to \mathbb{R}$, where $F_w(\mathbf{X})$ shall satisfy the following constraints: $\mathbb{E}_{\mathbf{X}\sim\mathbb{P}^T(\mathbf{X})}[F_w(\mathbf{X})] = 1$, and $\mathbb{E}_{\mathbf{X}\sim\mathbb{P}^S(\mathbf{X})}[1/(F_w(\mathbf{X}))] = 1$. By definition of the Radon-Nikodym derivative, these constraints must be satisfied.

Building on (5), we solve for the following upper bound in Theorem 2:

$$\max_{\theta(F_w)} \mathcal{R}^S + \epsilon \times \mathbb{E}_{\mathbb{P}^T(\mathbf{X})}\left[e^{(-F_w(\mathbf{X}))}\mathcal{H}(\hat{Y}|\mathbf{X})\right]$$
$$\text{s.t. } \mathbb{E}_{\mathbf{X}\sim\mathbb{P}^T(\mathbf{X})}[F_w(\mathbf{X})] = 1, \ \mathbb{E}_{\mathbf{X}\sim\mathbb{P}^S(\mathbf{X})}[1/(F_w(\mathbf{X}))] = 1 \tag{6}$$

Finally, we plug in the empirical risk estimator for $\mathcal{R}^S$, approximate the expectation in second term with the empirical version over $\mathcal{D}^T$, posit $\epsilon$ as a hyperparameter and add the unfairness objective in eq 3 to minimize the following:

$$\min_{\theta(F)} \max_{\theta(F_w)} \mathcal{L}(\theta(F), \theta(F_w)) = \widehat{ER}^S + \lambda_1 \frac{1}{m}\sum_{\mathbf{X}_i \in \mathcal{D}^T}\left[e^{(-F_w(\mathbf{X}_i))}\mathcal{H}(\hat{Y}|\mathbf{X})\right] + \lambda_2\hat{\mathcal{L}}_{Wass}(\mathcal{D}^T)$$

$$\text{s.t. } \mathcal{C}_1 = \frac{1}{m}\sum_{\mathbf{X}_i \in \mathcal{D}^T}F_w(\mathbf{X}_i) = 1, \ \text{and } \mathcal{C}_2 = \frac{1}{n}\sum_{\mathbf{X}_i \in \mathcal{D}^S}\frac{1}{F_w(\mathbf{X}_i)} = 1 \tag{7}$$

Here $\lambda_1$ and $\lambda_2$ are hyperparameters governing the objectives. $\mathcal{C}_1$ and $\mathcal{C}_2$ refer to the constraints.

---

**Algorithm 1:** Gradient Updates for the proposed objective to learn fairly under covariate shift

---

**Input:** Training data $\mathcal{D}^S$, Unlabelled Test data $\mathcal{D}^T$, model $F$, weight estimator $F_w$, decaying
learning rate $\eta_t$, number of pre-training steps $\tilde{\mathcal{E}}$, number of training steps $\mathcal{E}$ for eq 8,
$\lambda_1, \lambda_2$
**Output:** Optimized parameters $\theta^*(F)$
1 $\theta^0(F) \leftarrow$ random initialization
2 **for** $t \leftarrow 1$ **to** $\tilde{\mathcal{E}}$ **do**
3 $\quad \lfloor \quad \theta^t(F) \leftarrow \theta^{t-1}(F) - \eta_t \nabla_{\theta^{t-1}(F)} \widehat{ER}^S$
4 $\theta^{\tilde{\mathcal{E}}}(F_w) \leftarrow$ random initialization
5 **for** $t \leftarrow \tilde{\mathcal{E}} + 1$ **to** $\mathcal{E} + \tilde{\mathcal{E}}$ **do**
6 $\quad \mid \quad \theta^t(F_w) \leftarrow \theta^{t-1}(F_w) + \eta_t \nabla_{\theta^{t-1}(F_w)} \mathcal{L}(\theta^{t-1}(F), \theta^{t-1}(F_w))$ ; subject to $\mathcal{C}_1$ and $\mathcal{C}_2$
7 $\quad \mid \quad \theta^t(F) \leftarrow \theta^{t-1}(F) - \eta_t \nabla_{\theta(F)} \mathcal{L}(\theta^{t-1}(F), \theta^t(F_w))$; /* We apply gradient stopping through
$\quad \quad \lfloor \quad F_w$ during backpropagation in this step */
8 $\theta^*(F) \leftarrow \theta^{\mathcal{E} + \tilde{\mathcal{E}}}(F)$

---

Since the function has a representation layer followed by a classifier, i.e. $F = h \circ g$, in our implementation we apply the weighing function on $g(\cdot)$. Therefore, we have the following formulation:

$$\min_{\theta(F)} \max_{\theta(F_w)} \mathcal{L}(\theta(F), \theta(F_w)) = \widehat{ER}^S + \lambda_1 \frac{1}{m} \sum_{\mathbf{X}_i \in \mathcal{D}^T} \left[ e^{(-F_w(g(\mathbf{X}_i)))} \mathcal{H}(\hat{Y}|\mathbf{X}) \right] + \lambda_2 \hat{\mathcal{L}}_{Wass}(\mathcal{D}^T)$$

$$\text{s.t. } \mathcal{C}_1 = \frac{1}{m} \sum_{\mathbf{X}_i \in \mathcal{D}^T} F_w(g(\mathbf{X}_i)) = 1, \text{ and } \mathcal{C}_2 = \frac{1}{n} \sum_{\mathbf{X}_i \in \mathcal{D}^S} \frac{1}{F_w(g(\mathbf{X}_i))} = 1 \quad (8)$$

We use alternating gradient updates to solve the above min-max problem. Our entire learning procedure consists of *two stages*: (1) pre-training $F$ for some epochs with only $\mathcal{D}^S$ and (2) further training $F$ with (8). The procedure is summarized in Algorithm 1 and a high level architecture is provided in Figure 2.

## 6 EXPERIMENTS

We demonstrate our method on 4 widely used benchmarks in the fairness literature, i.e. Adult, Communities and Crime, Arrhythmia and Drug Datasets with detailed description in appendix A.2. The baseline methods used for comparison are: MLP, Adversarial Debias (AD) (Zhang et al., 2018), Robust Fair (RF) (Mandal et al., 2020), Robust Shift Fair (RSF) (Rezaei et al., 2021) and Z-Score Adaptation (ZSA) with detailed description in appendix A.3. The implementation details of all the methods with relevant hyperparameters are provided in section A.5. The procedure for constructing the covariate shift is described in section A.4. To summarize, we use the Principal Component Analysis (PCA) direction to generate covariate shifted test set similar to Rezaei et al. (2021); Gretton et al. (2008). The evaluation of our method against the baselines is done via the trade-off between fairness violation (using $\Delta_{EOdds}$) and error (which is $100-$ accuracy). All algorithms are run 50 times before reporting the mean and the standard deviation in the results.

### 6.1 COMPARATIVE RESULTS

The experimental results for the shift constructed using procedure in section A.4 are shown in Figure 3. The results closer to the *bottom left* corner in each plot are desirable. In some cases, the standard deviation bars in the figure stretch beyond 0 in $\mathbb{R}^-$ due to skewness when we plot standard error bars, however all the numbers across the runs are *positive*.

Our method provides better error and fairness tradeoffs against the baselines on all the benchmarks. For example, on the Adult dataset, we have the lowest error rate at around 15% with $\Delta_{EOdds}$ at almost 0.075 while the closest baselines MLP and RF fall short on either of the metrics. On Arrhythmia and Communities, our method achieves very low $\Delta_{EOdds}$ (best on Arrhythmia with a

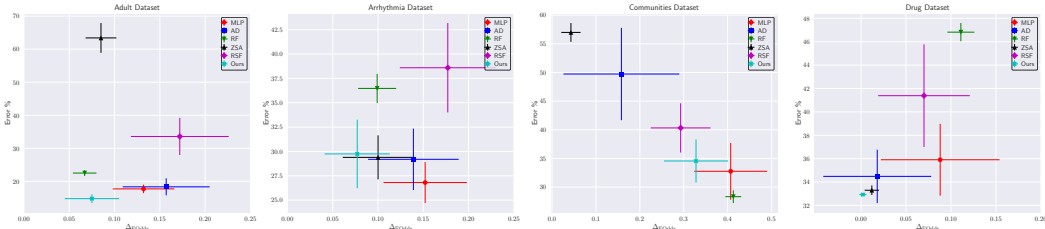

Figure 3: Comparison of our method against the baselines under Covariate Shift. The bars provide the standard deviation intervals both error (vertical) and $\Delta_{\text{EOdds}}$ (horizontal).

margin of $\sim 30\%$) with only marginally higher error as compared to MLP and RF respectively. On the Drug dataset, we achieve the best numbers for both the metrics. For the same accuracy, we obtain 1.3x-2x improvements against the baselines methods on most of the benchmarks. Similarly for the same $\Delta_{\text{EOdds}}$, we achieve up to 1.5x lower errors. It is also important to note that all the other unsupervised adaptation algorithms perform substantially worse and are highly unreliable. For example, ZSA performs well only on the Drug dataset, but shows extremely worse errors (even worse than *random predictions*) on Communities and Adult. The adaptation performed by ZSA is insufficient to handle covariate shift. RSF baseline is consistently worse across the board. This is because it tries to explicitly estimate $\mathbb{P}^S(\mathbf{X})$ and $\mathbb{P}^T(\mathbf{X})$ which is extremely challenging whereas we implicitly estimate the importance ratio. While there is extensive evidence in the literature suggesting that fairness is achieved at the expense of performance (Menon & Williamson, 2018; Zhao, 2021; Zliobaite, 2015), we attribute the low errors achieved by our method to the novel entropy formulation, where we in fact minimize the *worst case* weighting of entropy under the constraints. The saddle-point solution optimizes the entropy on points far from the training distribution, via appropriate scaling (importance weighting).

## 6.2 RESULTS ON ASYMMETRIC SHIFT

We also study the problem of covariate shift under a new lens where the degree of shift is substantially different across the groups, which also motivates our novel formulation (section 4).

To construct this, we follow the same procedure as described in section A.4, but operate on data for the two groups differently. The shift is introduced in one of the groups while for the other group, we resort to splitting it randomly into train-val-test. Figure 4 provides the results for the setup when shift is created in group A = 0 whereas figure 5 provides the result for shift in group A = 1.

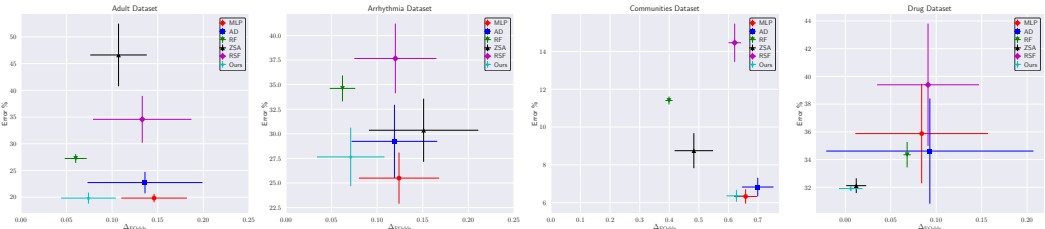

Figure 4: Comparison of our method against the baselines under asymmetric covariate Shift for group A = 0.

We again observe that our method provides better tradeoffs across the board. For the shift in group A = 0, we have substantially better results on Adult and Arrhythmia with up to $\sim 2$x improvements on $\Delta_{\text{EOdds}}$ for similar error and up to $\sim 1.4$x improvements in error for similar $\Delta_{\text{EOdds}}$. On the Communities dataset, MLP and AD show similar performance to ours, but much worse on the Drug dataset for both the metrics. ZSA performs comparably to our method only on Drug, but is substantially worse on other datasets. This confirms the inconsistency of the baselines under this setup as well. For the shift in group A = 1, we observe a similar behavior. On the Drug dataset, we clearly obtain the best tradeoff compared to all other baselines. MLP and AD achieve similar performance to our method on Communities, but show up to 2x worse $\Delta_{\text{EOdds}}$ on Arrhythmia with marginal improvements in error. On the Adult dataset, we observe up to 1.5x improvements against

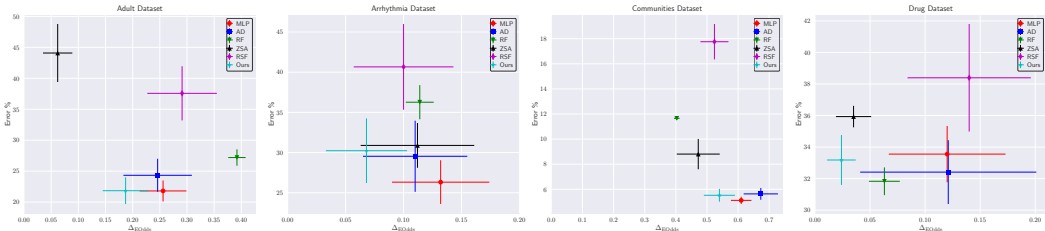

Figure 5: Comparison of our method against the baselines under asymmetric covariate Shift for group A = 1.

MLP and AD in $\Delta_{\text{EOdds}}$. RF baseline performs strictly worse than ours on Adult and Arrhythmia datasets where it's marginally better on either metrics on Communities and Drug, but at the expense of the other metric. It is also important to note that the errors are lower for all the methods as compared to figure 3 since only one group exhibits substantial shift while degradation in equalized odds is higher. This is in line with the reasoning provided in section 4 based on theorem 1.

## 6.3 RATIO ESTIMATED VIA $F_w(g(\mathbf{X}))$

We empirically justify the use of $F_w(g(\mathbf{X}))$ by comparing the distribution of the learned ratio across samples from $\mathcal{D}^T$ and $\mathcal{D}^S$ in figure 6.

It is evident that the parametrized weight network can approximately learn importance ratios w.r.t $\mathbb{P}^T$ and $\mathbb{P}^S$. The ratio computed for the test points lie mostly between 0 and 1 in order to satisfy $\mathcal{C}_1$ (in eq 8) whereas the ratio computed for the training points are mostly $> 1$ in order to satisfy $\mathcal{C}_2$ (in eq 8). More importantly, $F_w$ is learned end to end via optimization and doesn't incur any significant overhead compared to explicit density estimation.

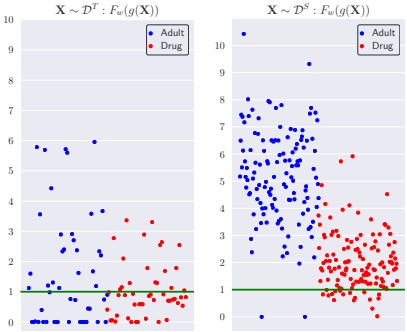

Figure 6: Comparison of the ratio estimated via $F_w(g(\mathbf{X}))$ across $\mathcal{D}^T$ and $\mathcal{D}^S$. The network learns the importance ratios w.r.t $\mathbb{P}^T$ and $\mathbb{P}^S$.

## 6.4 EXTENDED ANALYSIS

Extensive experimental results and analysis across multiple settings are provided in Appendix (due to lack of space). We empirically justify the motivation for unsupervised adaptation (described in section 4) in section A.6.1. Ablation studies for the hyperparameters $\lambda_1$ and $\lambda_2$ are performed in section A.6.2, for the magnitude of shift in section A.6.3 and for value of $m$ in section A.6.4. In section A.6.5 we derive the connection to standard entropy loss over unlabeled test samples (akin to the work by Wang et al. (2021a)) and demonstrate that our formulation achieves substantially better trade-off.

## 7 CONCLUSION

In this work, we considered the problem of unsupervised test adaptation under covariate shift to achieve good fairness-accuracy trade-offs when a small amount of unlabeled test data is available. We showed how fair representation matching alone is insufficient due to covariate shift. We proposed a composite objective that involves weighted entropy loss on the unsupervised test and a representation matching loss across protected groups. Finally, we experimentally demonstrate that our composite objective outperforms many baselines on benchmarks in achieving non trivial accuracy-fairness trade-offs.

## 8 REPRODUCIBILITY STATEMENT

We have described all the relevant implementation details required to reproduce the experiments in the appendix. The details for all the benchmarks as well as the baselines are provided comprehensively. We will publicly release the source code after the review process.

## 9 ETHICS STATEMENT

This work aims to address the concerns related to the unfairness and bias issues that manifest when there is a shift in distribution across training and the testing phase of a model. With the ever increasing real-world deployment of machine learning models, especially in life-altering scenarios like jurisdiction and college admissions, we hope to tackle these issues with this work and expect a cumulative social gain.

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

# A APPENDIX

## A.1 PROOFS

*Proof of Theorem 2.* We start with rewriting the expected cross entropy loss on the test as importance sampled loss on the training distribution.

$$\mathcal{R}^T = \mathbb{E}_{\mathbb{P}^T(\mathbf{X})} \left\{ \sum_{y \in \{0,1\}} -\mathbb{P}^T(\mathbf{Y} = y|\mathbf{X}) \log(P(\hat{\mathbf{Y}} = y|\mathbf{X})) \right\} \tag{9}$$

$$\overset{a}{=} \mathbb{E}_{\mathbb{P}^S(\mathbf{X})} \left\{ \left( \frac{d\mathbb{P}^T(\mathbf{X})}{d\mathbb{P}^S(\mathbf{X})} \right) \sum_{y \in \{0,1\}} -\mathbb{P}^S(\mathbf{Y} = y|\mathbf{X}) \log(P(\hat{\mathbf{Y}} = y|\mathbf{X})) \right\} \tag{10}$$

$(a)$ is the **Importance Weighting** technique proposed by (Sugiyama et al., 2007b) and $\frac{d\mathbb{P}^T(\mathbf{X})}{d\mathbb{P}^S(\mathbf{X})}$ is the Radon-Nikodym derivative because the two distributions are absolutely continuous with respect to each other.

$$= \mathcal{R}^S + \mathbb{E}_{\mathbb{P}^S(\mathbf{X})} \left( \frac{d\mathbb{P}^T(\mathbf{X})}{d\mathbb{P}^S(\mathbf{X})} - 1 \right) \left\{ \sum_{y \in \{0,1\}} -\mathbb{P}^S(\mathbf{Y} = y|\mathbf{X}) \log(P(\hat{\mathbf{Y}} = y|\mathbf{X})) \right\} \tag{11}$$

$$= \mathcal{R}^S + \mathbb{E}_{\mathbb{P}^T(\mathbf{X})} \left( 1 - \frac{d\mathbb{P}^S(\mathbf{X})}{d\mathbb{P}^T(\mathbf{X})} \right) \left\{ \sum_{y \in \{0,1\}} -\mathbb{P}^T(\mathbf{Y} = y|\mathbf{X}) \log(P(\hat{\mathbf{Y}} = y|\mathbf{X})) \right\} \tag{12}$$

$$\overset{b}{\leq} \mathcal{R}^S + \epsilon \times \mathbb{E}_{\mathbb{P}^T(\mathbf{X})} \left\{ \left( 1 - \frac{d\mathbb{P}^S(\mathbf{X})}{d\mathbb{P}^T(\mathbf{X})} \right) \sum_{y \in \{0,1\}} -P(\hat{\mathbf{Y}} = y|\mathbf{X}) \log(P(\hat{\mathbf{Y}} = y|\mathbf{X})) \right\} \tag{13}$$

$$\overset{c}{\leq} \mathcal{R}^S + \epsilon \times \mathbb{E}_{\mathbb{P}^T(\mathbf{X})} \left[ e^{\left( -\frac{d\mathbb{P}^S(\mathbf{X})}{d\mathbb{P}^T(\mathbf{X})} \right)} \mathcal{H}(\hat{\mathbf{Y}}|\mathbf{X}) \right], \tag{14}$$

$(b)$ is because of the assumption that $\frac{\mathbb{P}^T(\mathbf{Y}|\mathbf{X})}{P(\hat{\mathbf{Y}}|\mathbf{X})} \leq \epsilon$ almost surely with respect to $\mathbf{X} \sim \mathbb{P}^T$.

$(c)$ This is because $1 - x \leq e^{-x}, \; x \geq 0$. $\qquad\square$

## A.2 DATASET DESCRIPTION

The detailed description of the datasets used in this work are as follows:

- **Adult** is a dataset from the UCI repository containing details of individuals. The output variable is the indicator of whether the adult makes over \$50k a year. The group attribute is gender. Following Mandal et al. (2020), we use the processed data with 2213 examples and 97 features.
- **Arrhythmia** is a dataset from the UCI repository where each example is classified between the presence and absence of cardiac arrhythmia. The group attribute is gender. Following Rezaei et al. (2021), we used the dataset used containing 452 examples and 279 features.
- **Communities and Crime** is a dataset from the UCI repository where each example represents a community. The output variable is the community having a violent crime rate in the $70^{th}$ percentile of all the communities. The group attribute is the binary indicator of the presence of the majority white population. Following Mandal et al. (2020), we use the dataset with 2185 examples and 122 features.
- **Drug** is a dataset from the UCI repository where the task is to classify the type of drug consumer based on personality and demographics. The group attribute is race. Following Rezaei et al. (2021), we used the dataset with 1885 samples and 11 features.

### A.3 BASELINES

We use the following baselines for comparison. This covers the exhaustive set of relevant methods described in section 2.

- **MLP** is the standard *Multi Layer Perceptron* classifier that doesn't take into account shift and fairness properties. In the standard in-distribution evaluation settings, such a model usually provides the upper bound to the accuracy without considering fairness, however the scenario differs as we are dealing with distribution shifts.

- **Adversarial Debiasing (AD)** (Zhang et al., 2018) is one of the most popular debiasing methods in the literature. This method performs well on the fairness metrics under the standard in-distribution evaluation settings, but fails to do so in the shift setting.

- **Robust Fair (RF)** (Mandal et al., 2020) proposes a framework to learn classifiers that are fair not only with respect to the training distribution, but also for a broad set of distributions characterized by any arbitrary weighted combinations of the dataset.

- **Robust Shift Fair (RSF)** (Rezaei et al., 2021) is a recent and most relevant baseline to this work. The authors propose a method to robustly learn a classifier under covariate shift, with fairness constraints. A severe limitation of this method is that it requires explicit estimation of both source and target covariates' distributions.

- **Z-Score Adaptation (ZSA)** following the thread of work under *Batch Norm Adaptation* (Li et al., 2017; Schneider et al., 2020) literature, we implement a baseline that adapts the parameters of the normalizing layer by recomputing the z-score statistics from the unlabeled test data points.

### A.4 SHIFT CONSTRUCTION

To construct the covariate shift in the datasets, i.e., to introduce $\mathbb{P}^S(\mathbf{X}, \mathbf{A}) \neq \mathbb{P}^T(\mathbf{X}, \mathbf{A})$, we utilize the following strategy akin to the works of Rezaei et al. (2021); Gretton et al. (2008). First, all the non-categorical features are normalized by *z-score*. We then obtain the *first principal component* of the of the covariates and further project the data onto it, denoting it by $\mathcal{P}_\mathcal{C}$. We assign a score to each point $\mathcal{P}_\mathcal{C}[i]$ using the density function $\Xi : \mathcal{P}_\mathcal{C}[i] \to \mathrm{e}^{\gamma \cdot (\mathcal{P}_\mathcal{C}[i] - b)}/\mathcal{Z}$. Here, $\gamma$ is a hyperparameter controlling the level of distribution shift under the split, $b$ is the $60^{th}$‰ (percentile) of $\mathcal{P}_\mathcal{C}$ and $\mathcal{Z}$ is the normalizing coefficient computed empirically. Using this, we sample $40\%$ instances from the dataset as the test and remaining $60\%$ as training. To construct the validation set, we further split the training subset to make the final train:validation:test ratio as $5 : 1 : 4$, where the test is distribution shifted.

Note that for large values of $\gamma$, all the points with $\mathcal{P}_{\mathcal{C}[i]} > b$ will have high density thereby increasing the probability of being sampled into the test set. This generates a sufficiently large distribution shift. Correspondingly, for smaller values of $\gamma$, the probability of being sampled is not sufficiently high for these points thereby leading to higher overlap between the train and test distributions.

### A.5 IMPLEMENTATION DETAILS

We use the same model architecture across MLP and our method in order to ensure consistency. Following Wang et al. (2021b), a *Fully Connected Network* (FCN) with $4$ layers is used, where the first two layers compose $g$ and the subsequent layers compose $h$. For AD, we use an additional $2$ layer FCN that serves as the *adversarial head* $a : g(\mathbf{X}) \to \mathcal{A}$ (similar to (Wang et al., 2021b)).

Without further specification, we use the following hyperparameters to train MLP, AD and ZSA. The number of epochs is set to $50$ with *Adam* as the optimizer (Kingma & Ba, 2014) and weight decay of $1e^{-5}$ (for Adult dataset, the weight decay is $5e^{-4}$). The learning rate is set to the value of $1e^{-3}$ initially and is decayed to $0$ using the *Cosine Annealing* scheduler (Loshchilov & Hutter, 2017). A batch size of $32$ is generally used to train the models. The gradients are clipped at the value of $5.0$ to avoid explosion during training. The dropout (Srivastava et al., 2014) rate is set to $0.25$ across the layers. For AD, the adversarial loss hyperparameter post grid search is used.

RF and RSF works have tuned their model for the specific architecture and corresponding hyperparameters (different from the aforementioned specifics). We perform another grid search over these hyperparameters and report the best results for comparison.

For our proposed method, we pre-train the model for 15 epochs with only $\widehat{ER}^S$. For the next 35 epochs we use the objective in eq 8, but with a higher training data batch size to reduce variance in the *Monte Carlo Estimation* of the second constraint $\left( \frac{1}{n} \sum_{\mathbf{X}_i \in \mathcal{D}^S} \frac{1}{F_w(g(\mathbf{X}_i))} = 1 \right)$. The value of $m$ (size of $\mathcal{D}^T$) is kept at 50 for the main experiments, which is $<<$ size of $\mathcal{D}^S$. The primary experiments are run with the shift magnitude $\gamma = 10$ (with ablations provided in section A.6.3). The constraints $\mathcal{C}_1$ and $\mathcal{C}_2$ as mentioned in 8 are implemented as squared error terms where we minimize $c_1 \cdot \left( \left( \frac{1}{m} \cdot \sum_{\mathbf{X}_i \in \mathcal{D}^T} F_w(g(\mathbf{X}_i)) \right) - 1 \right)^2 + c_2 \cdot \left( \left( \frac{1}{n} \cdot \sum_{\mathbf{X}_i \in \mathcal{D}^S} \frac{1}{F_w(g(\mathbf{X}_i))} \right) - 1 \right)^2$, where $c_1$ and $c_2$ are hyperparameters to control the relative importance of each constraint. The values of the tuple $(\lambda_1, \lambda_2)$ are set to the following - Adult : $(1, 0.01)$ ; Arrhythmia : $(0.01, 0.005)$ ; Communities : $(0.005, 0.0001)$ and Drug : $(0.1, 0.1)$ post grid search. All experiments are run on single NVIDIA Tesla V100 GPU.

## A.6 ANALYSIS

### A.6.1 UNSUPERVISED ADAPTATION WITH OUR ENTROPY FORMULATION UNDER ASYMMETRIC SHIFT

The asymmetric shift setup described in section 3.1 provides a well grounded motivation (section 4) and use case for explicitly handling shifts along with the unfairness objective. We complement the claim with empirical evidence here. The results in table 1 provide comparison of the performance across the metrics with and without our proposed formulation. The wasserstein objective in eq 3 is retained in both settings. We observe significant improvements on both error and $\Delta_{\text{EOdds}}$ with our formulation. Particularly on the Drug dataset, we see an improvement of almost $4\%$ in error and around $13\times$ in the $\Delta_{\text{EOdds}}$, which is also notable on Arrhythmia.

Table 1: Comparison of the performance on using the unfairness objective without and with the unsupervised adaptation (our proposed entropy formulation). We observe substantial improvements in both error and $\Delta_{\text{EOdds}}$. Numbers in the parenthesis represent standard deviation across the 50 runs.

| Dataset | Arrhythmia | | Drug | |
|---|---|---|---|---|
| Entropy Variation | Without Entropy | With Entropy (eq 8) | Without Entropy | With Entropy (eq 8) |
| Error % | 28.648 (3.079) | 27.617 (2.978) | 35.859 (3.437) | 31.910 (0.186) |
| $\Delta_{\text{EOdds}}$ | 0.080 (0.032) | 0.071 (0.037) | 0.076 (0.043) | 0.006 (0.013) |

### A.6.2 VARIATION OF $\lambda_1$ AND $\lambda_2$

In this section, we study the variation of the performance of our method against the hyperparameters governing error ($\lambda_1$) and $\Delta_{\text{EOdds}}$ ($\lambda_2$). While studying the effect of either, we keep the other constant.

Table 2: Variation of the performance of our method with Entropy Regularizer $\lambda_1$ on Adult dataset.

| $\lambda_1 \rightarrow$ | 0 | 0.001 | 0.005 | 0.01 | 0.1 | 1.0 |
|---|---|---|---|---|---|---|
| Error (in %) | 23.819 (8.593) | 22.047 (6.631) | 20.510 (6.706) | 20.851 (7.829) | 14.626 (1.318) | 14.787 (1.326) |
| $\Delta_{\text{EOdds}}$ | 0.131 (0.038) | 0.126 (0.037) | 0.129 (0.037) | 0.129 (0.029) | 0.104 (0.033) | 0.075 (0.30) |

Table 2 reports the variation for $\lambda_1$ keeping $\lambda_2 = 0.01$ fixed. It is evident from the numbers that increasing $\lambda_1$ has strong correlation with the reduction in error, which exhibits a saturation at $0.1$. Higher values of $\lambda_1$ emphasize the minimization of the worst-case weighted entropy thus helping in calibration of the network in regions across $\mathbb{P}^T$. Furthermore, we observe significant improvements in $\Delta_{\text{EOdds}}$ which is inline with the motivation of handling shifts along with an unfairness objective (section 4). Increasing $\lambda_1$ doesn't help post a threshold value as the correct estimation of

the true class for a given $\mathbf{X}$ under $\mathbb{P}^T$ becomes harder, particularly in regions far from the labeled in-distribution data. Imposing very strong $\lambda_1$ can hurt the model performance.

The variation against $\lambda_2$, keeping $\lambda_1 = 1$ fixed is reported in table 3. As $\lambda_2$ increases, we observe a gradual improvement in $\Delta_{\text{EOdds}}$. This exhibits a maxima after which the performance degrades drastically. This is because strongly penalizing $\hat{\mathcal{L}}_{Wass}(\mathcal{D}^T)$ with a small number of samples $m$ leads to overfitting (illustrated by the large standard deviation) while matching $\mathbb{P}^T(\mathbf{X}|\mathbf{A})$. This also hurts the optimization as demonstrated by the substantial increase in error.

Table 3: Variation of the performance of our method with Wasserstein Regularizer $\lambda_2$ on Adult dataset.

| $\lambda_2 \rightarrow$ | 0 | 0.001 | 0.005 | 0.01 | 0.1 | 1.0 |
|---|---|---|---|---|---|---|
| Error (in %) | 15.049 (1.424) | 15.849 (1.437) | 14.901 (1.352) | 14.787 (1.326) | 17.936 (15.962) | 42.280 (32.581) |
| $\Delta_{\text{EOdds}}$ | 0.091 (0.031) | 0.099 (0.034) | 0.098 (0.032) | 0.075 (0.030) | 0.074 (0.036) | 0.093 (0.064) |

### A.6.3 SHIFT MAGNITUDE

We study the variation of the performance of our method against the magnitude of shift $\gamma$ on Arrhythmia. A comparison against the best baseline ZSA is also provided. The variation of error is plotted in the left subfigure of 7. With no shift in the data, $\gamma = 0$, we observe that both the methods exhibit small errors as $\mathcal{D}^T$ follows in-distribution. With the increase in the value of $\gamma$, ZSA shows a sudden increment in the error with an unstable pattern whereas our method exhibits a more gradual pattern and lower error as compared to ZSA. This justifies that the weighted entropy objective helps.

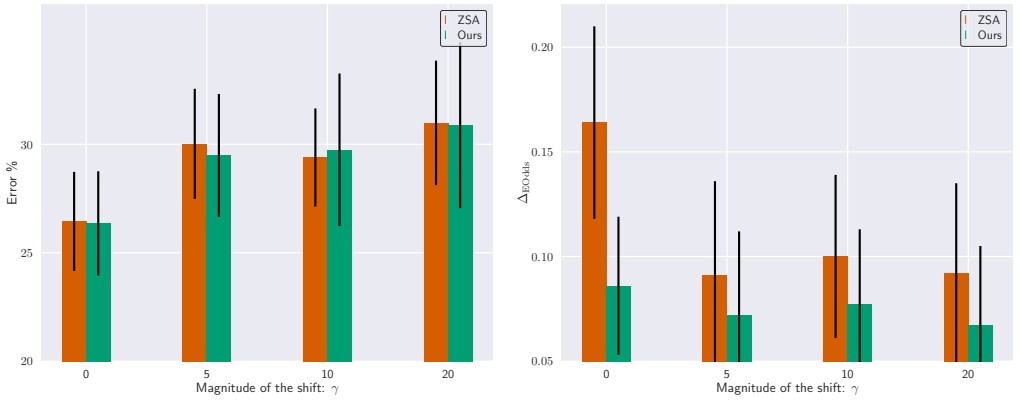

Figure 7: Variation of error against $\gamma$ (left subfigure) and $\Delta_{\text{EOdds}}$ against $\gamma$ (right subfigure) on Arrhythmia Dataset. We observe that our method performs better in both metrics against the best baseline ZSA. While the error increases gradually, but we observe substantially better $\Delta_{\text{EOdds}}$ for our method.

On the contrary, we observe that our method is highly stable over $\Delta_{\text{EOdds}}$ and performs consistently better for larger shifts as compared to ZSA. We attribute this effect to the proposed objective which optimizes the model to learn fairly under the shift and over the worst case scenario.

### A.6.4 VARIATION OF SIZE OF $\mathcal{D}^T$

Here, we study the dependence of the methods on the size of $\mathcal{D}^T$. The left subfigure in 8 plots the variation of error against $m$. The error gradually decreases for our method and RSF as the estimation of the true test distribution improves and the optimization procedure covers a larger region of $\mathbb{P}^T$. This also makes the approximation by $F_w$ much more reliable and closer to true ratios. Although, the results don't show notable improvements after a certain threshold as we are dealing in

an unsupervised regime over $\mathbb{P}^T$. It becomes increasingly harder to correctly estimate the true class for a given $\mathbf{X}$ under $\mathbb{P}^T$, particularly in regions far from the labeled in-distribution data. Interestingly, ZSA doesn't exhibit any improvements which demonstrates that merely matching first and second order moments across the data is not sufficient to handle covariate shifts.

The right subfigure in 8 plots the variation of $\Delta_{\text{EOdds}}$ against $m$. Here, we observe a consistent reduction in $\Delta_{\text{EOdds}}$ as more data from $\mathbb{P}^T$ helps is matching representations via improved approximation of $\mathbb{P}^T(\mathbf{X}|A)$. Further this objective only deals with matching representations across the groups and doesn't stagnate as quickly with increasing $m$ as the error margins, which suffers from lack of reliable estimation in regions far from in-distribution.

We consistently outperform RSF in both very small and larger regimes of $m$, partly verifying the importance of $F_w$ rather than a direct estimation of $\mathbb{P}^S$ and $\mathbb{P}^T$ as RSF does. ZSA is substantially worse than both RSF and our method in terms of errors. In terms of $\Delta_{\text{EOdds}}$ its only marginally better than our method for $m = 10$ and $m = 20$, but at a huge expense of prediction performance.

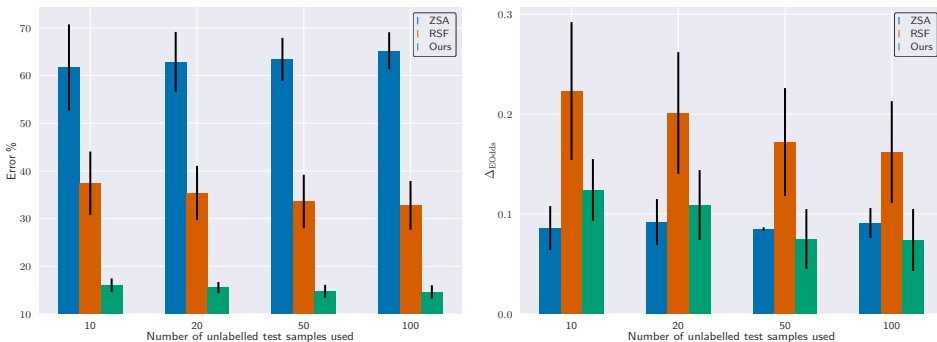

Figure 8: Variation of error against $m$ (left subfigure) and $\Delta_{\text{EOdds}}$ against $m$ (right subfigure) on Adult Dataset. We see reduction in both error and $\Delta_{\text{EOdds}}$ with increasing value of $m$.

A.6.5 UNWEIGHTED ENTROPY VS OUR WEIGHTED ENTROPY FORMULATION

It is easy to observe that we can recover the standard unlabeled test entropy minimization using our derivation. Formally specifying, we can upper bound eq 14 to obtain entropy as follows:

$$\mathcal{R}^S + \epsilon \times \mathbb{E}_{\mathbb{P}^T(\mathbf{X})}\left[e^{\left(-\frac{d\mathbb{P}^S(\mathbf{X})}{d\mathbb{P}^T(\mathbf{X})}\right)}\mathcal{H}(\hat{Y}|\mathbf{X})\right] < \mathcal{R}^S + \epsilon \times \mathbb{E}_{\mathbb{P}^T(\mathbf{X})}\left[\mathcal{H}(\hat{Y}|\mathbf{X})\right], \because e^{-x} \leq 1, \forall x \geq 0$$

(15)

Our formulation particularly provides a tighter bound as compared to standard entropy and implicitly accounts for points in $\mathcal{D}^T$ that are close to $\mathbb{P}^S$ by assigning low weight. The experimental results comparing the two settings both with and without the unfairness objective are provided in table 4. Our formulation achieves substantially better results with a relative improvement of around 33% in error. Note that due to the fairness-error tradeoff, the standard (unweighted) entropy achieves better $\Delta_{\text{EOdds}}$, but that is achieved at the expense of a nearly random classifier as evident from the error rate of nearly 50%. We also highlight the large standard deviation in the results achieved by unweighted entropy. This is largely because it seeks to minimize entropy across all $m$ points whereas our objective is more adaptive based on the approximation of importance ratio.

Table 4: Comparison of the performance of Standard Unweighted Entropy v/s our Weighted Entropy formulation on Communities dataset.

| | Without Wasserstein Objective (eq 3) | | With Wasserstein Objective (eq 3) | |
|---|---|---|---|---|
| Entropy Variation → | Unweighted Entropy | Weighted Entropy (Ours) | Unweighted Entropy | Weighted Entropy (Ours) |
| Error % | 45.787 (12.900) | 34.291 (4.463) | 45.654 (13.090) | 35.549 (3.748) |
| $\Delta_{\text{EOdds}}$ | 0.204 (0.194) | 0.359 (0.074) | 0.201 (0.200) | 0.328 (0.073) |

### A.6.6 COMPARISON TO OTHER DENSITY RATIO ESTIMATION METHODS

As the explicit computation of the density values can be hard, we estimate the ratio $\frac{d\mathbb{P}^S(\mathbf{X})}{d\mathbb{P}^T(\mathbf{X})}$ via a parametrized network for this class of baselines. Density ratio estimation methods were previously proposed in the works by Sugiyama et al. (2007c) (KLIEP), Kanamori et al. (2009) (LSIF). Menon & Ong (2016) analysed these methods in a unifying framework.

To experimentally demonstrate the efficacy of our method over the aforementioned, we use the density ratio estimation methods of KLIEP and LSIF in the following manner. First, the importance ratio $\frac{d\mathbb{P}^S(\mathbf{X})}{d\mathbb{P}^T(\mathbf{X})}$ is estimated using unsupervised test samples and the training samples available based on the KLIEP and LSIF losses (given in Menon & Ong (2016)) via a parameterized weight network $s(X)$. Then, we train a classifier based on the following instance weighted cross entropy loss and representation matching loss:

$$\min_{\theta(F=h\circ g)} \frac{1}{n} \sum_{(\mathbf{X}_i, \mathbf{Y}_i, A_i)\in\mathcal{D}^S} s(\mathbf{X}_i)\left(-\log P_{\theta(F)}(\hat{\mathbf{Y}}=Y_i|\mathbf{X}_i)\right) + \lambda\hat{\mathcal{L}}_{Wass}(\mathcal{D}^T) \tag{16}$$

where $s(X)$ is a non-negative function which is obtained by minimizing:

$$L_{\text{KLIEP}}(s(\mathbf{X})) = \frac{1}{m}\sum_{\mathbf{X}_i\in\mathcal{D}^T} -\log s(\mathbf{X}_i) + \left(\frac{1}{n}\sum_{\mathbf{X}_i\in\mathcal{D}^S} s(\mathbf{X}_i) - 1\right)^2 \tag{17}$$

or,

$$L_{\text{LSIF}}(s(\mathbf{X})) = \frac{1}{m}\sum_{\mathbf{X}_i\in\mathcal{D}^T} -s(\mathbf{X}_i) + \frac{1}{2}\left(\frac{1}{n}\sum_{\mathbf{X}_i\in\mathcal{D}^S} (s(\mathbf{X}_i))^2\right) \tag{18}$$

The results are stated in tables 5 and 6. First, we observe that our method consistently outperforms these algorithms across the datasets. The relative improvement of our method is as high as $\sim 31\%$ in error on Adult dataset and $\sim 32.5\times$ in $\Delta_{\text{EOdds}}$ on Drug dataset against LSIF. Similar non-trivial margins can be noted on other datasets. Second, the variance in accuracies of the KLIEP and LSIF based importance is very high on the Drug dataset. Particularly, both KLIEP and LSIF exhibit up to $20-40$ times higher variance in error and up to $10-12$ times in $\Delta_{\text{EOdds}}$.

**Key Takeaway:** We can attribute this to the phenomenon that in the small sample regime, importance weighted training on training dataset alone may not bring any improvements for covariate shift due to variance issues and thus estimating the ratio can be insufficient. In fact, this has been pointed out in Menon & Ong (2016). We, on the other hand propose a new formulation to optimize for an upper bound based on the ratio estimation but due to the negative exponent of the importance ratio, the importance ratio's effect on the loss does not induce such high variance and it also leverages unsupervised test samples at training time.

Table 5: Comparison of our method against popular density ratio estimation methods: KLIEP and LSIF on Drug and Adult datasets.

| Dataset | Drug | | Adult | |
|---------|------|---|-------|---|
| Method | Error % | $\Delta_{\text{EOdds}}$ | Error % | $\Delta_{\text{EOdds}}$ |
| KLIEP | 34.782 (3.879) | 0.043 (0.042) | 20.787 (4.368) | 0.124 (0.028) |
| LSIF | 39.517 (5.944) | 0.065 (0.049) | 19.428 (3.203) | 0.107 (0.028) |
| Ours | **32.928** (0.143) | **0.002** (0.004) | **14.787** (1.326) | **0.075** (0.030) |

### A.6.7 FAIRNESS-ERROR TRADEOFF CURVES

To further demonstrate the effectiveness of our method, we plot the *Pareto Frontier* in figure 9 (variance bars are removed to retain clarity), similar to Agarwal et al. (2018). Achievable trade-offs for the baselines are plotted along with our Pareto curve for comparison. We observe that the curve corresponding to our method is closer to the left axis with a high gradient. The error reduces drastically for a small increase in $\Delta_{\text{EOdds}}$ while providing better tradeoffs as compared to the *optimal performance* of the baselines.

Table 6: Comparison of our method against popular density ratio estimation methods: KLIEP and LSIF on Communities and Arrhythmia datasets.

| Dataset | Communities | | Arrhythmia | |
|---|---|---|---|---|
| Method | Error % | $\Delta_{\text{EOdds}}$ | Error % | $\Delta_{\text{EOdds}}$ |
| KLIEP | 38.466 (4.403) | 0.323 (0.062) | 30.630 (3.605) | 0.085 (0.044) |
| LSIF | 38.842 (3.518) | 0.340 (0.055) | 30.972 (3.467) | 0.081 (0.042) |
| Ours | **34.549** (3.748) | **0.328** (0.073) | **29.746** (3.519) | **0.077** (0.036) |

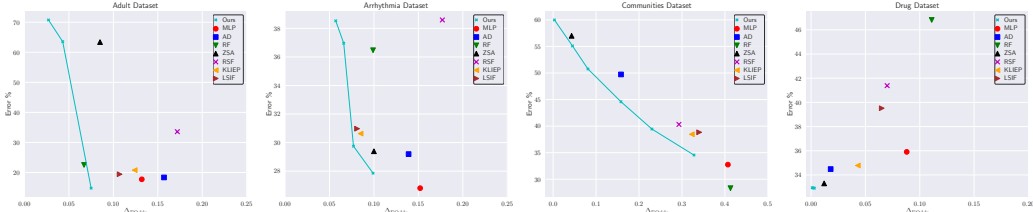

Figure 9: Fairness-Error Tradeoff Curves for our method (Pareto Frontier) against the optimal performance of the baselines. Our method provides better tradeoffs in all cases. (On Drug dataset, the performance is concentrated around the optimal point). Variance bars are removed to retain clarity in the plots.

### A.6.8 EMPIRICAL INVESTIGATION OF THE BOUND $\epsilon$ IN THEOREM 2

We compare the ratio of the prediction probabilities for the classes ($y \in \{0, 1\}$) on the validation set (which is not available during training to our algorithm) between a classifier trained only on the training set (Train) and a classifier trained only on the held-out test set (Test).

We plot the ratios in figure 10 with outliers removed. The subfigures (a),(b) demonstrate the ratio for the true class label for the samples. Subfigures (c),(d) demonstrate the ratio for class $y = 0$ and subfigures (e),(f) demonstrate the ratio for class $y = 1$. Correspondingly, in figure 11 we plot the ratios with outliers. Note that atmost **4** points in every plot are outliers with $ratios > 5$. This empirically justifies that $\epsilon$ can be set *not too high* with high probability except for a few outliers.

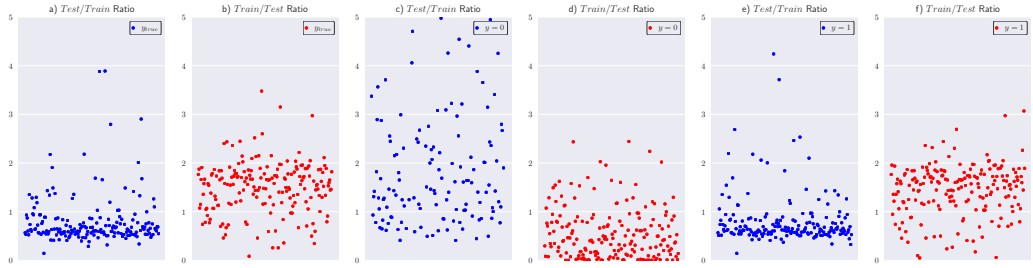

Figure 10: The subfigures demonstrate the ratio of the prediction probabilities for the classes ($y \in \{0, 1\}$) on the validation set between a classifier trained only on the training set (Train) and a classifier trained only on the held-out test set (Test), with outliers removed. Note that $\epsilon = 5$ provides a reasonable threshold and holds for all the samples but for 4 outliers (shown in figure 11).

### A.6.9 COMPARISON OF ACCURACY PARITY

We further demonstrate that our method is better as compared to the baselines when the fairness metric is *Accuracy Parity*. The results for Adult dataset are provided in table 7.

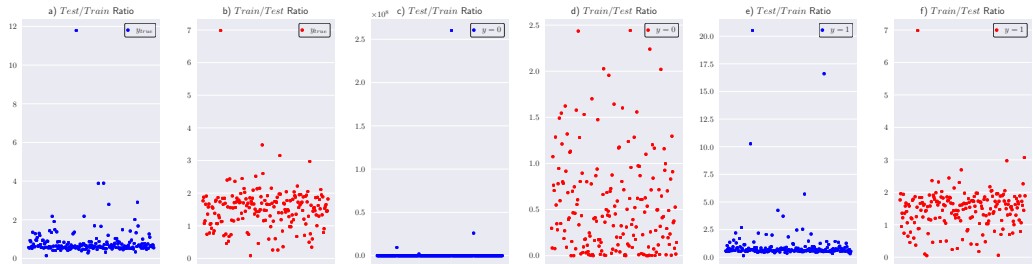

Figure 11: The subfigures demonstrate the ratio of the prediction probabilities for the classes ($y \in \{0,1\}$) on the validation set between a classifier trained only on the training set (Train) and a classifier trained only on the held-out test set (Test), with outliers. Atmost **4** points in every plot are outliers with ratios $> 5$.

Table 7: Comparison of Accuracy Parity as well as Error for all the methods. We outperform the baselines, particularly KLIEP and LSIF that are prone to poor results due to high variance.

| Method -> | MLP | AD | RF | RSF | ZSA | KLIEP | LSIF | Ours |
|---|---|---|---|---|---|---|---|---|
| Error % | 17.735 | 18.356 | 22.525 | 33.591 | 63.396 | 20.787 | 19.428 | **14.787** |
| Accuracy Parity % | 5.764 | 3.156 | 6.094 | 11.851 | 11.131 | 4.417 | 6.677 | **2.990** |

### A.7 COMPARISON OF GENERALIZATION BOUNDS BETWEEN IMPORTANCE SAMPLED TRAINING LOSS AND OUR OBJECTIVE

In this section, we would like contrast Generalization bounds between our objective (Right Hand Side of 5) and the left hand side which is importance sampled training loss.

The main intention is to bring out dependence on the variance of importance ratios. Therefore, we make the following simplifying assumptions:

**Assumption 3.**
- *Let $\Theta = \{\theta_1 \ldots \theta_k\}$ be parameters of a finite set of classifiers of the form $P_\theta(\hat{Y}|\mathbf{X})$.*

- *Let us assume the loss function $\ell_1(Y, \mathbf{X}; \theta) = -\log P_\theta(Y|\mathbf{X})$ is bounded between $[0,1]$ in the domain $\{0,1\} \times \mathcal{X}$ for all $\theta \in \Theta$.*

- *Let the loss function $\ell_2(\mathbf{X}; \theta) = \sum_{y \in \{0,1\}} -P_\theta(\hat{Y} = y|\mathbf{X})\log P_\theta(\hat{Y} = y|\mathbf{X})$ be also be bounded between $[0,1]$ in the domain $\{0,1\} \times \mathcal{X}$ for all $\theta \in \Theta$.*

- *Let us assume we have access to the exact importance weight $w(\mathbf{X}) = \frac{\mathbb{P}^T(\mathbf{X})}{\mathbb{P}^S(\mathbf{X})}$. Since, we assume $\mathbb{P}^T(\cdot)$ and $\mathbb{P}^S(\cdot)$ are absolutely continuous with respect to each other, $w(\mathbf{X}) > 0, \forall \mathbf{X} \in \mathcal{X}$. For convenience of notation, let $\tilde{w}(\mathbf{X}) = \frac{\mathbb{P}^S(\mathbf{X})}{\mathbb{P}^T(\mathbf{X})}$.*

- *Let $\sup_{\mathbf{X} \in \mathcal{X}} w(\mathbf{X}) = M$. Let the variance of the importance ratio with respect to the training distribution be $\mathbb{E}_{\mathbf{X} \sim \mathbb{P}^S}[w^2(\mathbf{X})] = \sigma^2$.*

**Remark:** We have assumed $\ell_1$ is bounded in $[0,1]$. If the log loss over a suitable function class is Lipschitz and domain is bounded, then the loss is also bounded. Therefore, it is not a very heavy assumption and we wanted to keep the analysis simple and normalized.

There are two loss functions we compare:

1. $R_{IS}(\theta) = \sum_{(\mathbf{X}_i, Y_i) \sim \mathcal{D}^S} w(\mathbf{X}_i)\ell_1(Y_i, \mathbf{X}_i; \theta)$ and

2. $R_{WE}(\theta) = \sum_{(\mathbf{X}_i, Y_i) \sim \mathcal{D}^S} \ell_1(Y_i, \mathbf{X}_i; \theta) + \lambda \sum_{(\mathbf{X}_i, Y_i) \sim \mathcal{D}^T} e^{-\tilde{w}(\mathbf{X}_i)}\ell_2(Y_i, \mathbf{X}_i; \theta)$.

$R_{IS}(\theta)$ is the empirical importance sampled loss while $R_{WE}$ is the weighted entropy objective of Theorem 2. We recall some generalization bounds for finite hypothesis classes with bounded risks.

**Definition 3.** Rademacher complexity $\mathcal{R}(A)$ for a finite set $A = \{a_1, a_2 \ldots a_N\} \subset \mathbb{R}^n$ is given by:

$$\mathcal{R}(A) = \mathbb{E}_\sigma[\sup_{a \in A} \sum_i \sigma_i a[i]] \tag{19}$$

where $\sigma$ is a sequence of $n$ i.i.d Rademacher variables each uniformly sampled from $\{-1, -1\}$ and $a[i]$ is the $i$-th coordinate of vector $a$.

*Empirical Rademacher complexity* of a class of finite number of functions $\mathcal{F}$ on a data set $\mathcal{D}$ with $m$ samples is given by $\mathcal{R}(\mathcal{F}(\mathcal{D}))$ where $\mathcal{F}(\mathcal{D}) = \{\text{vec}(f(x), \forall x \in D), \forall f \in \mathcal{F}\}$. Here, $\text{vec}(\cdot)$ is a vector of entries.

**Theorem 4** (Bousquet et al. (2003)). $\mathcal{R}(\mathcal{F}(\mathcal{D})) \leq \left( \sup_{f \in \mathcal{F}, x \in \mathcal{D}} |f(x)| \right) \sqrt{\frac{2 \log |\mathcal{F}|}{|\mathcal{D}|}}.$

**Theorem 5.** *[Bousquet et al. (2003)] When a dataset $\mathcal{D}$ is sampled i.i.d from distribution $\mathbb{P}(\mathbf{X})$ and $f$ is uniformly bounded by $L$ over the domain of $\mathbb{P}$, then with probability $1 - \delta$ over the draw of $\mathcal{D}$,*

$$\mathbb{E}_{x \sim \mathbb{P}}[f(x)] \leq \frac{1}{|\mathcal{D}|} \sum_{x \in \mathcal{D}} f(x) + 2\mathcal{R}(\mathcal{F}(\mathcal{D})) + 3L\sqrt{\frac{\ln(2/\delta)}{2|\mathcal{D}|}}$$

$$\leq \frac{1}{|\mathcal{D}|} \sum_{x \in \mathcal{D}} f(x) + 2L\sqrt{\frac{2 \log |\mathcal{F}|}{|\mathcal{D}|}} + 3L\sqrt{\frac{\ln(2/\delta)}{2|\mathcal{D}|}}, \forall f \in \mathcal{F} \tag{20}$$

**Theorem 6.** *Under Assumption 3, we have that with probability $1 - 2\delta$ over the draws of $\mathcal{D}^S \sim \mathbb{P}^S$ and $\mathcal{D}^T \sim \mathbb{P}^T$, we have $\forall \theta \in \Theta$*

$$\mathbb{E}_{\mathbb{P}^S, \mathbb{P}^T}[R_{WE}(\theta)] \leq R_{WE}(\theta) + 2\sqrt{\frac{2 \log |\Theta|}{|\mathcal{D}^S|}} + 2\lambda\sqrt{\frac{2 \log |\Theta|}{|\mathcal{D}^T|}} + 3\sqrt{\frac{\ln(2/\delta)}{2|\mathcal{D}^S|}} + 3\lambda\sqrt{\frac{\ln(2/\delta)}{2|\mathcal{D}^T|}} \tag{21}$$

*Proof.* We apply Theorem 5 to $\ell_1(\cdot)$ (which is bounded by 1) and $e^{-\tilde{w}(\cdot)}\ell_2(\cdot)$ where $\ell_2(\cdot) \leq 1$, $e^{-\tilde{w}(\cdot)} \leq 1$ with the appropriate datasets in Assumption 3. We then use union bound over the two error events that result from application of the theorem twice. $\square$

For finite hypothesis classes, we recall generalization bounds for importance sampled losses from Cortes et al. (2010b).

**Theorem 7** (Cortes et al. (2010b)). *Suppose that a dataset $\mathcal{D}$ is sampled i.i.d from distribution $\mathbb{P}(\mathbf{X})$, $f$ is uniformly bounded by $L$ over the domain of $\mathbb{P}$, and a fixed weighing function $w(x)$ is such that $\sup w(x) = M$, $\mathbb{E}_{x \sim \mathbb{P}}[w(x)^2] \leq \sigma^2$. Consider the loss function $\tilde{f}(x) = w(x)f(x)$. We denote $\tilde{f} = w \circ f$. then with probability $1 - \delta$ over the draw of $\mathcal{D}$, we have:*

$$\mathbb{E}_{x \sim \mathbb{P}}[\tilde{f}(x)] \leq \sum_{x \in \mathcal{D}} \tilde{f}(x) + \frac{2M(\log |\Theta| + \log(1/\delta))}{3\mathcal{D}^S} + L\sqrt{2\sigma^2 \frac{(\log |\mathcal{F}| + \log(1/\delta))}{|\mathcal{D}|}},$$

$$\forall \tilde{f} \in \{w \circ f, f \in \mathcal{F}\} \tag{22}$$

Applying Theorem 7 to $R_{IS}(\theta)$ we have the following result.

**Theorem 8.** *Under Assumption 3, we have that with probability $1 - \delta$ over the draws of $\mathcal{D}^S \sim \mathbb{P}^S$, we have $\forall \theta \in \Theta$*

$$\mathbb{E}_{\mathbb{P}^S}[R_{IS}(\theta)] \leq R_{IS}(\theta) + \frac{2ML(\log |\Theta| + \log(1/\delta))}{3|\mathcal{D}^S|} + \sqrt{2\sigma^2 \frac{(\log |\Theta| + \log(1/\delta))}{|\mathcal{D}^S|}} \tag{23}$$

*Proof.* The proof is a direct application of Theorem 7 to $R_{IS}(\theta)$ under Assumption 3. $\square$

**Key Takeaways:** Comparing Theorem 6 and Theorem 8, we see that the generalization bounds for importance sampled training loss depends on variance of importance ratio and also the worst ratio over the training set ($M$ and $\sigma^2$). In contrast, our objective *does not* depend on these parameters primarily due to negative exponential dependence on $\tilde{w}$. We also note that $R_{WE}$ depends on size of test set also while the other does not seem to. However, $R_{IS}$ needs to estimate importance ratios - which will depend on the test set . We have analyzed both losses when the importance ratios are assumed to be known just to bring out the difference in dependencies on other parameters.

**Remark:** In Assumption 3, we have assumed a finite hypothesis class $\Theta$. However, our result for Theorem 6 would generalize (as is) with rademacher complexity or covering number based arguments of infinite functions classes $\ell_1$ and $\ell_2$. Cortes et al. (2010b) also point out analogous generalization for the importance sampling loss.

