# OpenReview forum: "Unsupervised Adaptation for Fairness under Covariate Shift"
_ICLR.cc/2023/Conference — Submitted to ICLR 2023_

### Official Review · Reviewer_iDgW · 2022-10-22

**Confidence:** 3
**Correctness:** 3
**Technical Novelty And Significance:** 3
**Empirical Novelty And Significance:** Not applicable
**Recommendation:** 8

**Clarity, Quality, Novelty And Reproducibility:**

The paper was written in a clear way for others to reproduce. The formulation is also novel in the sense that they combine a weighted entropy objective (to address covariate shift) with representation matching (to address fairness). Overall I am satisfied with the quality of the paper.

**Strength And Weaknesses:**

Overall this is a well-written paper with solid theoretical results and comprehensive empirical findings.

Strength:
1. Novel formulation to address covariate shift along with fairness
2. Solid theoretical results to support the formulation
3. Comprehensive empirical results to further support the formulation
4. Thorough literature review for relevant works

Weakness:
I have several questions that I hope the authors can further clarify.
1. The new proposed method utilizes the test set information to address covariate shift. In some scenarios the test set is not known a priori. I am wondering in the experiments section, whether the baselines that the authors compared to, use the test set information or not. If they did not use the test set information, can we call it a fair comparison?
2. Furthermore, when considering the baseline choices, have you considered any of the methods mentioned in Section 2: Fairness under Distribution shift? I feel like those methods are more relevant to what you studied in this paper. I am interested in seeing the performance comparison with those methods if possible.
3. The representation matching bears a similar spirit with the paper: Shalit, U., Johansson, F.D. and Sontag, D., 2017, July. Estimating individual treatment effect: generalization bounds and algorithms. In International Conference on Machine Learning (pp. 3076-3085). PMLR, where they also tried to minimize a distributional distance between representations in two groups. I am curious why you chose the Wasserstein metric, even though Theorem 1 suggests the total variation distance.
4. In Section 5.1, why do we maximize (instead of minimize) over \theta in Eq. (6)?

**Summary Of The Paper:**

This paper considered the problem of unsupervised test adaptation under covariate shift to achieve good fairness-accuracy trade-offs when a small amount of unlabeled data is available. The authors proposed a new weighted entropy based loss function to account for covariate shift, in combination with a representation matching term to address fairness. A set of empirical experiments were conducted on four popular datasets to show the performance of their proposed method in achieving non trivial accuracy-fairness trade-offs.


**Summary Of The Review:**

As mentioned above, I saw several strengths of this paper:

1. Novel formulation to address covariate shift along with fairness
2. Solid theoretical results to support the formulation
3. Comprehensive empirical results to further support the formulation
4. Thorough literature review for relevant works

I also like the connection they made between a weighted entropy and importance sampling.

---

### Official Review · Reviewer_DiLH · 2022-10-25

**Confidence:** 3
**Correctness:** 3
**Technical Novelty And Significance:** 3
**Empirical Novelty And Significance:** 3
**Recommendation:** 5

**Clarity, Quality, Novelty And Reproducibility:**

The paper is generally clearly written. Codes are not attached but there are enough details.

The technical sections are solid however the novelty is limited. Also how fairness is really related to the new weighted entropy is not clear.

**Strength And Weaknesses:**

Strength

Fairness under the covariate shift is an important under-investigated topic and very relevant to the community.

The empirical results are significant.

Weakness

The lack of the target labels means the proposed method cannot be applied to all the fairness metrics that requires the knowledge of the true labels. So the metric considered in the paper is only an accuracy disparity. This will significantly affect the applicability of the method. Also, in the experimental results, I think the x-axis should be changed as the metric is not the real equalized odds.

The main contribution of the proposed formulation is actually in the first term, which is the covariate shift correction term, not the fairness regularization term. So it seems the resulting formulation is a weighted entropy loss for covariate shift correction (ratio estimation) in a mini-max end-to-end training. The novelty seems to be limited. For example, without fairness regularization, we can also apply the first term to learning problems under covariate shift. The application of weighted entropy under covariate shift is also not new. How are the fairness constraints connected to the covariate shift?



**Summary Of The Paper:**

This paper focuses on the fair learning problem under the covariate shift. The main contribution is the weighted entropy loss component of the regularized objective function. In the experiments, the proposed method achieves a better tradeoff between accuracy and fairness.

**Summary Of The Review:**

I am leaning towards rejection as the novel contribution to fair learning is limited, even though the weighted entropy is an effective method under the covariate shift.

---

### Official Review · Reviewer_HTwH · 2022-10-28

**Confidence:** 4
**Clarity, Quality, Novelty And Reproducibility:** Details are mentioned in the Strength…
**Correctness:** 2
**Technical Novelty And Significance:** 2
**Empirical Novelty And Significance:** 2
**Recommendation:** 3

**Strength And Weaknesses:**

The authors study an important question, which generalizes standard fairness issues to distribution shift scenarios (in particular, covariate shift). I value the importance of this setting. However, I think the methods have several unclear parts and the experiments should also be improved.

For the method part:
1. The first main issue is the unclear fairness notion throughout the paper. Equalized odds and accuracy parity correspond to two different fairness targets and there are different methods to deal with them. The authors should highlight the main fairness notion they target and show why their method could guarantee or approximate the fairness notion.
2. Theorem 2 needs further explanation. Firstly, the assumption on the constant $\epsilon$ is unclear. The value of $\epsilon$ depends on the model parameter and I think it can easily tend to infinity if a model outputs a small $P(\hat{Y}=y|X)$. Secondly, the superiority of the proposed theorem compared with Theorem 1 (Zhao & Gordon (2019)) is unclear. As I mentioned before, I think the value of $\epsilon$ may be large in general settings and the second term in the RHS of Theorem 2 may also be large, making the bound loose. The authors should discuss how Theorem 2 could guarantee handling the asymmetric covariate shift setting.
3. The comparison with density ratio estimation methods is unclear, such as the methods proposed in [1]. The authors claim that "the typical way of density estimation in high dimensions is particularly hard". However, Equation (5) requires the estimation of density ratio only and several methods could deal with the problem (such as the KLIEP and LSIF loss mentioned in [1]). As a result, it would be better if the authors can compare their algorithm with these methods.
4. I am unclear why we need the max step to optimize the density ratio function. The $F_w(X)$ should be the density ratio between the training and test distribution and the authors are encouraged to demonstrate why the max step could lead to the estimation of the true density ratio.

For the experimental part:
1. As mentioned before, the authors should compare the methods that use typical density ratio estimation methods [1] to estimate the function $F_w(X)$ instead of the max step.
2. The fairness-accuracy trade-off curves are encouraged to be plotted. See examples in [2].

Other minor typos:
1. Throughout the paper: upto -> up to
2. Second paragraph in the introduction: Similarly, (missing a comma here)
3. The cross-entropy loss in Equation (1) is wrong. The equations miss the $Y_i$ term.

[1] Menon, Aditya, and Cheng Soon Ong. "Linking losses for density ratio and class-probability estimation." International Conference on Machine Learning. PMLR, 2016.

[2] Agarwal, Alekh, et al. "A reductions approach to fair classification." International Conference on Machine Learning. PMLR, 2018.

**Summary Of The Paper:**

The authors study the fairness issue under the covariate shift setting. To address this problem, they first show that the previous bound in (Zhao & Gordan (2019)) can not handle the asymmetric covariate shift setting. Afterward, they propose a novel bound on the performance in the test distribution. They further develop a min-max optimization framework to calculate the proposed bound. Finally, they conduct experiments on several datasets to prove the effectiveness of their method.

**Summary Of The Review:**

Although the authors study an important problem, the methods have several unclear parts and the experiments should also be improved. As a result, I vote for the rejection in this round.

---

### Decision · Program_Chairs · 2023-01-20

**Decision:**

Reject

**Justification For Why Not Higher Score:**

Although the authors have made a lot of efforts to clarify things in the rebuttal period, the committee feels that the paper can be in a better shape and can make a much higher impact after some major modifications to clarify the method and points/concerns raised in the rebuttal discussion.

The reviewers still have concerns about the theory, which means that the theorems can be further clarified in a modified version of the manuscript.  It is hard for the reviewers to verify/justify the added theorems during a post-review discussion period.

Some baselines in experiments are missing. For example, the ones listed in An et al. (2022). Overall, the results are good but some results are not too convincing yet, such as the Pareto Frontier Curves.

Overall, the committee is looking forward to seeing a revision of this paper as a strong submission in the next venue.

**Justification For Why Not Lower Score:**

N/A

**Metareview: Summary, Strengths And Weaknesses:**

The paper considers a very important problem of adaptation of fairness under covariate shift.

Although the authors have made a lot of efforts to clarify things in the rebuttal period, the committee feels that the paper can be in a better shape and can make a much higher impact after some major modifications to clarify the method and points/concerns raised in the rebuttal discussion.

The reviewers still have concerns about the theory, which means that the theorems can be further clarified in a modified version of the manuscript.  It is hard for the reviewers to verify/justify the added theorems during a post-review discussion period.

Some baselines in experiments are missing. For example, the ones listed in An et al. (2022). Overall, the results are good but some results are not too convincing yet, such as the Pareto Frontier Curves.

Overall, the committee is looking forward to seeing a revision of this paper as a strong submission in the next venue.